# A survey of proximal methods for monitoring leaf phenology in temperate deciduous forests

Kamel Soudani[1], Nicolas Delpierre[1], Daniel Berveiller[1], Gabriel Hmimina[2], Jean-Yves Pontailler[1], Lou Seureau[1], Gaëlle Vincent[1], Éric Dufrêne[1]

[1] Université Paris-Saclay, CNRS, AgroParisTech, Ecologie Systématique et Evolution, 91405, Orsay, France
[2] Laboratoire de Météorologie Dynamique, IPSL, CNRS/UPMC, Paris, France

*Correspondence to:* Kamel Soudani (kamel.soudani@universite-paris-saclay.fr)

**Abstract.** Tree phenology is a major driver of forest-atmosphere mass and energy exchanges. Yet, tree phenology has rarely been monitored in a consistent way throughout the life of a flux tower site. Here, we used seasonal time-series of ground-based NDVI (Normalized Difference Vegetation Index), RGB camera GCC (Greenness Chromatic Coordinate), broad-band NDVI, LAI (Leaf Area Index), $f$APAR (fraction of Absorbed Photosynthetic Active Radiation), CC (Canopy Closure), $f$R$_{vis}$ (fraction of Reflected Radiation) and GPP (Gross Primary Productivity) to predict six phenological markers detecting the start, middle and end of budburst and of leaf senescence in a temperate deciduous forest using an asymmetric double sigmoid function (ADS) fitted to time-series. We compared them to observations of budburst and leaf senescence achieved by field phenologists over a 13-year period. GCC, NDVI and CC captured the interannual variability of spring phenology very well ($R^2 > 0.80$) and provided the best estimates of the observed budburst dates, with a mean absolute deviation (MAD) less than 4 days. For the CC and GCC methods, mid-amplitude (50%) threshold dates during spring phenological transition agreed well with the observed phenological dates. For the NDVI-based method, on average, the mean observed date coincides with the date when NDVI reaches 25% of its amplitude of annual variation. For the other methods, MAD ranges from 6 to 17 days. ADS method used to derive the phenological markers provides the most biased estimates for the GPP and GCC. During the leaf senescence stage, NDVI- and CC-derived dates correlated significantly with observed dates ($R^2 = 0.63$ and 0.80 for NDVI and CC, respectively), with MAD less than 7 days. Our results show that proximal sensing methods can be used to derive robust phenological metrics. They can be used to retrieve long-term phenological series at EC flux measurement sites and help interpret the interannual variability and trends of mass and energy exchanges.

## 1 Introduction

In the temperate and boreal climate zone, the timing of phenological events is strongly controlled by temperature and is thus responsive to the ongoing climate change (Menzel et al. 2006; Badeck et al. 2004; Piao et al. 2019). The opening of buds ("budburst") in spring and the coloration and fall of leaves ("leaf senescence") in autumn are the key steps in the phenological cycle of forest trees. These stages mark the start and end of the photosynthetically active period and as such

strongly influence the carbon and water exchanges between the ecosystem and the atmosphere (Goulden et al. 1996; Delpierre et al. 2009a; Richardson et al. 2010; Dragoni et al. 2011). Historically, the timing of these events has been monitored through direct and periodic human-eye observations of the state of buds and leaves in the field (Sparks and Carey, 1995). However, this method is time-consuming, laborious, and subject to an observer effect (Roetzer et al. 2000; Schaber and Badeck, 2002;

Klosterman et al. 2014). Alternative, ground-based indirect methods have been tested for monitoring the phenology of different ecosystems. Proximal sensing methods based on measuring the radiation reflected, transmitted or absorbed by the canopy (henceforth 'radiation-based methods') are increasingly being used. Broad-band NDVI calculated from measurements of the fraction of reflected radiation in the Photosynthetically Active Radiation (PAR) spectral domain and shortwave bands, proposed by Huemmrich et al. (1999) has been successfully used in order to monitor vegetation phenology in many studies

(Huemmrich et al. 1999; Richardson et al. 2007; Liu et al. 2019). Forest phenology was also described from measurements of the fraction of transmitted PAR through the canopy (Toda and Richardson, 2018; Perot et al. 2019) and Leaf Area Index (LAI) (Keenan et al. 2014). Spectral vegetation indices derived from tower-mounted hyperspectral spectroradiometers (Kobayashi et al. 2018; Lu et al. 2018), RGB/IR cameras (Richardson et al. 2007; Klosterman et al. 2014; Richardson et al. 2018; Richardson, 2019; Milliman et al. 2019) or from two bands red and near infrared proximal sensors (Ruy et al. 2010; Eklundh

et al. 2011; Soudani et al. 2012; Hmimina et al. 2013) have also been assessed. More recently, passive sun-induced fluorescence has been used (Lu et al. 2018). In vegetation sites where continuous measurements of carbon flux are available, phenology has also been estimated from the dynamics of GPP (gross primary productivity) and net ecosystem exchange (NEE) (Gonsamo et al. 2013; Wu et al. 2017; Garrity et al. 2011).

Over the past two decades, hundreds of experimental sites measuring $CO_2$, water and energy exchanges between ecosystems and the atmosphere have been set up worldwide. These sites are organized in networks (Fluxnet, ICOS, etc.) and aim to record long-term data according to standardized protocols (Baldocchi et al. 2001; Franz et al. 2018). These sites acquire high temporal resolution time-series combining both mass ($CO_2$ and water) flux data with ancillary data which include incident, reflected and transmitted radiation measurements in different spectral ranges, and also LAI, NDVI, and RGB images of the

canopy. Yet, the phenology of the vegetation cover is not routinely monitored over all sites, precluding the assessment of its influence on carbon and water exchanges. These sites provide data which allow the comparison of various radiation-based methods for monitoring forest phenology. However, the comparative studies cited above and those carried out at some of the carbon flux measurement sites did not cover all the methods on the same site and were also limited to a few and for short periods of time. Also, most of these studies suffered from a lack of direct and independent phenological observations. As

underlined in Klosterman et al. (2014), this is a key challenge in interpreting estimates from the various approaches. Indeed, most of the radiation-based methods use optical signals at different wavelengths and at different spectral resolutions. Depending on species and sensor specifications (spectral, radiometric and geometric responses), this could lead to possible mismatches between observed and estimated phenology due to the well-known selective absorption properties of plant components (Sims and Gamon, 2002). The measurement conditions (sun-view geometry, field of view) may also differ

(Sonnentag et al. 2012). Also, some mainly observe the top of the canopy (down-looking sensors mounted above the canopy) while others are more integrative of the whole canopy (indirect methods that use transmitted or absorbed radiation). Therefore, there is a need to conduct comparative studies to establish rigorously the correspondence between phenological dates recorded by field phenologists and phenological metrics predicted by indirect proximal methods.

In this study, we present an exhaustive comparative survey of various proximal methods to estimate both spring and autumn phenology in a mature deciduous forest ecosystem surrounding the Fontainebleau-Barbeau carbon flux tower. The main objective is to evaluate the performance of each of the methods in reproducing inter-annual variation of spring and autumn phenology directly observed by field phenologists, over a 13-year period.

## 2. Materials and Methods

**2.1. Site description**

Data were mainly acquired in the eddy-covariance (EC) flux measurement site at the Fontainebleau-Barbeau forest (48°28'26"N., 2° 46'57" E.), 53 km southeast of Paris, France. Fontainebleau-Barbeau is a deciduous forest mainly composed of mature sessile oak (*Quercus petraea* (Matt.) Liebl), and an understory of hornbeam (*Carpinus betulus* L.). The average stand LAI, based on measurements using litter collection method over 2012-2018 period, is 5.8 m²/m², ranging from 4.6 to 6.8
m²/m² (unpublished data). Hornbeam contribution to stand LAI accounts for 30%, ranging from 24% to 39% from year to year.

In Fontainebleau-Barbeau EC flux measurement site, which belongs to the European ICOS-RI Ecosystem network (Integrated Carbon Observation System - Research Infrastructure, FR-Fon code), a 35-m high tower was installed in 2005 in order to measure energy and $CO_2$ exchanges between the forest and the atmosphere with the eddy-covariance technique. More
details about the study site and flux calculation are given in Delpierre et al. (2016). The tower has been equipped with various proximal sensors that we used here to estimate the timings of phenological events (Table 1). More details about the instrumentation and measurements achieved in this site are available in www.barbeau.universite-paris-saclay.fr.

**2.2. Extraction of phenological markers**

Data and methods used in the calculation of phenology metrics are summarized in Table 1. The general principle of
the phenological metrics extraction method consists in building time-series at daily resolution that describe the canopy foliage dynamics during the whole seasonal cycle of vegetation (Figure 1). This method applies to all the variables ("Vegetation variable", *Vv*) listed in (Table 1). Then, to compare the different vegetation proxies without possible methodological biases, we opted for the same method using an asymmetric double sigmoid (ADS) similar to Zhang et al. (2003); Soudani et al. (2008); Klosterman et al. (2014).


Briefly, an asymmetric double sigmoidal function was fitted on $Vv$ time-series according to the following equation:

$$Vv(t) = (w_1 + w_2) + \frac{1}{2}(w_1 - w_2)[\tanh(w_3(t-u)) - \tanh(w_4(t-v))]$$                    Eq.1


$Vv$ (t) is the considered vegetation variable (% of open buds and % of non-senescent leaves, NDVI, NDVI$br$, $f$R$_{vis}$, $f$APAR, CC, LAI, GCC or GPP). $t$ is the time (day of year). $tanh$ is the hyperbolic tangent and $w_1$, $w_2$, $w_3$, $w_4$, $u$, $v$ are the fitting parameters. ($w_1$+$w_2$) is the $Vv$ minimum in unleafy season. ($w_1$-$w_2$) is the total amplitude of variation of $Vv$ over the year. The two phenological markers $u$ and $v$ are the dates of the two inflection points when $Vv$ increases during the spring ($u$) and

decreases during the autumn ($v$). For these two dates $u$ and $v$, $Vv$(t) is very close to 50% of its total amplitude of variation, in spring and autumn respectively. Four other phenological markers are determined numerically from the extrema of the third derivative of the ADS function according to Zhang et al. (2003) (Figure 1). The six phenological markers are named according to Klosterman et al. (2014) as follows: SOS, MOS and EOS for the start, middle, and end of leaf onset (budburst) in spring and SOF, MOF and EOF for the start, middle and end of leaf senescence in autumn, corresponding approximately to 10%,

50% and 90% of total amplitude during the increase and the decline in canopy greenness in spring and autumn, respectively.

Fitting was done by minimizing the sum of squares of differences between fitted (Eq.1) and measured $Vv$. In order to better constrain the fitting at the end of the leafy season, each year of data was extended to the end of January of the following year. Thus, potentially, each time-series is composed of 396 days instead of 365 days.

### 2.3. Data

**2.3.1. Field phenological observations (OBS)**

We collected spring and autumn phenological field observations at the Fontainebleau-Barbeau forest over 13 years (2006-2018; see Delpierre et al. 2020, Denéchère et al. 2019) through complementary sampling schemes. Over the 2006-2018 period, we implemented an 'extensive' survey in which we monitored bi-weekly over March-April the bud development of >100 randomly chosen dominant sessile oak trees, and recorded the date at which 50% of the individuals displayed at least

50% buds open in their crowns (corresponding to stage 7 of the BBCH scale). Observations were done with binoculars by three inter-calibrated observers. This date is referred to as BB-OBS (BB for budburst) in the following. In years 2015-2017 we complemented this protocol with an 'intensive' survey. Twenty-seven to 66 individual trees (depending on years) were tagged and monitored for bud burst from 0% budburst to 100% budburst in each tree crown. This survey yielded the progress of budburst for each tree crown, that we averaged to get the progress of budburst for the tree population (Figure 2a). We further

monitored the progress of leaf senescence (% of colored or fallen leaves) in each individual tree crown weekly in autumn, and

averaged the individual values to get the progress of leaf senescence at the tree population scale (Figure 2a). We fitted the ADS (eq. 1) function to these continuous data and retrieved the MOS-OBS (in spring) and MOF-OBS (in autumn) metrics. The MOS-OBS (obtained from the intensive survey) and BB-OBS (obtained from the extensive survey) dates compare very well, their maximum absolute difference being 1-day (Delpierre et al. 2020). Hence in the following we will use the BB-OBS as the observed date of budburst over the whole (2006-2018) study period. All spring phenological observations were conducted on a bi-weekly basis. Hence the uncertainty of BB-OBS is 3.5 days.

We completed the MOF-OBS (autumn) metrics obtained at Fontainebleau-Barbeau through the intensive survey over 2015-2017 with leaf senescence data obtained over 2011-2014 from a phenological survey site 50-km away from Fontainebleau-Barbeau (Orsay site). At this site, we deployed an intensive-monitoring protocol of leaf senescence (30 to 60 tagged sessile oaks monitored weekly for the percentage of colored or fallen leaves during autumn) from which we obtained the LS-OBS metrics, that is the date at which 50% trees had 50% leaves colored or fallen. In 2015, autumn phenological observations were conducted simultaneously in Fontainebleau-Barbeau and Orsay: the MOF-OBS (Fontainebleau-Barbeau, DoY 300) and LS-OBS (Orsay, DoY 295) dates compared well. Considering that leaf senescence dates are comparable at a scale of tens of kilometres (Delpierre et al. 2009b), we used the 2011-2014 Orsay LS-OBS data to complement the 2015-2017 Fontainebleau-Barbeau MOF-OBS data. All spring phenological observations were conducted on a weekly basis. Hence the uncertainty of MOF-OBS and LS-OBS is 7 days.

### 2.3.2. Narrow-band NDVI

The NDVI is calculated as follows:

$$NDVI = (NIR - R)/(NIR + R) \hspace{2cm} \text{Eq.2}$$

$R$ and $NIR$ are radiances in the red (640-660 nm) and the near infrared (780-920 nm) bands, respectively. Radiances are measured using a laboratory made NDVI sensor (Pontailler et al. 2003). A description of this sensor and its use for estimating phenological metrics in various biomes is given in Soudani et al. (2012) and Hmimina et al. (2013). Briefly, the sensor is positioned at the top of the EC flux tower in Fontainebleau-Barbeau forest, about 7 m above the canopy, directed downwards and inclined about 20-30° to the vertical and facing south to avoid the hot-spot effects in canopy reflectance when the viewing direction is collinear with the solar direction. The field of view of the sensor was 100° and the area observed is a few tens of m². Measurements are acquired continuously every half-hour. Noisy data, mainly due to rainfall and very low radiation conditions, were removed according the procedure described in Soudani et al. (2012). This procedure consists in keeping only NDVI measurements recorded when the ratio between global radiation ($R_{Gin}$) measured above the canopy and the exo-atmospheric radiation ($R_{ex}$) at the top of atmosphere exceeds the threshold of 0.6, considered to be the threshold for distinguishing between clear and overcast sky conditions (Soudani et al. 2012). Then, daily average of filtered NDVI data

acquired between 10h and 14h (UT) is considered to minimize the effects of daily variations in solar angle. Finally, filtered and daily averaged NDVI data were used in Eq.1.

### 2.3.3. RGB camera

Digital pictures (resolution of 2590 x 1920 pixels) of the forest canopy are acquired continuously every hour between 8h – 17h (UT) with an Axis P1347 camera installed next to and according to the same geometric configuration of the NDVI sensor. In order to minimize effects of changing illumination conditions, a white PVC panel is installed in the camera field of view (FOV) and used as a reference. Pictures (10/day) were processed automatically under MATLAB. At first, three regions of interest (ROI) were delineated on a spring picture. Two ROI, having an area of 3000 pixels and 1140 pixels, respectively, are located on the reference panel. The third ROI is located over the vegetation area that covers the central region of the picture (2 M pixels). To convert RGB data measured by the camera to pseudo-reflectance ($\rho R$, $\rho G$, $\rho B$), digital counts in Red, Green and Blue bands of the vegetation ROI were averaged and divided by the averages of R, G and B measured on the two white ROI on the reference PVC panel. These pseudo-reflectances were averaged on daily basis (10 values per day, corresponding to the hourly sampling) and used to determine daily **Greenness Chromatic Coordinate (GCC)** as follows:

$$GCC = \rho G/(\rho R + \rho G + \rho B) \qquad\qquad \text{Eq.3}$$

Phenological markers are then extracted from GCC time-series according Eq.1.

### 2.3.4. Broad-band NDVIbr and fraction of reflected radiation fRvis

Broad-band NDVI (NDVI*br*), named according to Huemmrich et al. (1999), was calculated from incoming and reflected radiation in the visible spectral region (400-700 nm) corresponding to the spectral range of PAR measured using PAR sensors (PQS1, Kipp and Zonen, Finland) and in the shortwave spectral regions (200 to 3600 nm) using a CMP22 pyranometer (Kipp and Zonen, Finland). A conversion factor of 4.57 µmol J$^{-1}$ (McCree, 1972 in Wang et al. 2006) was used to convert PAR unit (µmol m$^{-2}$ s$^{-1}$) to energy unit (J m$^{-2}$ s$^{-1}$). As in Wohlfart et al. (2010), NDVI*br* is calculated as below:

$$NDVIbr = \frac{\left(\frac{NIR_{out}}{NIR_{in}}\right)-\left(\frac{PAR_{out}}{PAR_{in}}\right)}{\left(\frac{NIR_{out}}{NIR_{in}}\right)+\left(\frac{PAR_{out}}{PAR_{in}}\right)} \qquad\qquad \text{Eq.4}$$

$NIR_{in} = RG_{in} - PAR_{in}$

$NIR_{out} = RG_{out} - PAR_{out}$

$RG_{in}$, $RG_{out}$, $PAR_{in}$, $PAR_{out}$ are incoming and outgoing reflected radiation in shortwave and PAR spectral regions.

The fraction of reflected radiation $fR_{vis}$ was calculated as:

$$fR_{vis} = \left(\frac{PAR_{out}}{PAR_{in}}\right)$$
Eq.5

NDVI*br* and *f*R*vis* were filtered by applying the same ratio of 0.6 between R$_{Gin}$ and R$_{ex}$ and limiting the period of acquisition between 10h to 14h TU. Finally, filtered and daily averaged *f*R*vis* and NDVI*br* data were used to in Eq.1 to extract the six phenological markers. Because *f*R*vis* was lower during the leafy season than in winter (unleafy season), Eq.1 was applied to (1-*f*R*vis*) allowing to have the same temporal pattern as the other variables. For simplicity, *f*R*vis* term will be used hereafter when
referring to the method.

### 2.3.5. Fraction of absorbed PAR fAPAR, Canopy Closure CC and Leaf Area Index LAI

Fifteen quantum PAR sensors (PQS1, Kipp and Zonen, Finland), directed towards the sky, are installed below canopy on the ground-area surrounding the EC flux tower to ensure a robust spatial sampling of the radiation transmitted through the canopy. Measurements are achieved at a half-hour time step, simultaneously with measurements of incoming and
reflected PAR radiation above the canopy. The filtering of transmitted, reflected and incoming radiation measurements is carried out according to the same procedure used for NDVI, NDVI*br* and *f*R*vis*. Consequently, only measurements taken between 10h and 14h TU after filtering are used in the calculation of *fAPAR*, CC and LAI.

*f*APAR is calculated according the following expression:

$$fAPAR = \frac{PAR_{in} - PAR_{out} - PAR_t}{PAR_{in}}$$
Eq.6

Canopy closure CC is calculated using a new formulation as follows:

$$CC = 1 - \left(\frac{PAR_t}{cos(\theta)}\right)/PAR_{in}$$
Eq.7

Where PAR$_{in}$ and PAR$_{out}$ are defined above in Eq.3. PAR$_t$ is the averaged over 15 sensors of transmitted radiation
measured beneath the canopy. $\theta$ is the sun zenith angle calculated using the standard astronomical formula. Unlike Eq. 6 and the previous studies (Richardson et al. 2007; Garrity et al. 2011; Toda and Richardson, 2018), the division of PAR$_t$ by the cosine of the sun zenith angle (Eq. 7) allows to consider variation of PAR$_t$ due solely to the variation of the path length of incident radiation passing through the forest canopy before reaching the ground according to the seasonal variation of the solar angle. In order to assess the performance of this new formulation proposed in this study, we also calculated CC without cosine
correction.

Another possible alternative to this correction/normalization in order to take into account sun angle effects on transmitted PAR (Eq.7) is to estimate Leaf Area Index from the canopy gap fractions since the estimation of LAI using Beer-Lambert law corrects for the effects of solar angle and considers leaf angle distribution through the extinction coefficient *K*. The LAI was calculated as follows:

$$LAI = -log\ (PAR_t/PAR_{in})/K$$
Eq.8

log is the natural logarithm. *K* is the coefficient of extinction, calculated following the expression given in Campbell and Norman (1998):

$$K(\theta) = \frac{\sqrt{x^2 + tan(\theta)^2}}{x + 1.774\,(x+1.182)^{-0.733}}$$
Eq.9

The parameter *x* describes an ellipsoidal leaf angle distribution function (*x*=1 for spherical distribution, *x* > 1 for planophile and *x* < 1 for erectophile leaves). In this study and in order to let *K* vary according to the seasonal variations of the solar angle, we only fixed the parameter *x* in Eq.9. In order to estimate an average value of *x* parameter in the Fontainebleau-

Barbeau forest, Eq.8 was inverted, based on direct LAI measurements around the EC flux tower using litter collection technique according to the ICOS protocol (Gielen et al. 2018) and the radiation measurements over 2012-2018 period. *x* was about 1.4 which corresponds to an average value of *K* of about 0.67 during the leafy season (DOY 150-240). This value agrees with previous studies (Baldocchi et al. 1984; Holst et al. 2004). Thus, we note that *K* is calibrated from the "true" average green LAI measured by the litter collection method, and thus it corrects for clumping effects and woody components. The term LAI

is used in the present study instead of the term PAI (Plant Area Index, including lead and woody components) usually used when it is estimated from canopy transmittance and using assumptions about leaf angle distribution in order to estimate the extinction coefficient (Campbell, 1986).

Similarly, to the other vegetation variables, phenological metrics were extracted from time-series of *f*APAR, CC

and LAI according Eq.1.

### 2.3.6. GPP data

Half-hourly GPP data were estimated on the ecosystem from net-carbon flux measurements acquired by an eddy covariance system. Details of instrumentation and processing are provided in Delpierre et al. (2016) and on www.barbeau.universite-paris-saclay.fr. GPP was aggregated daily and used to create continuous time-series from 2006 to

2018. Extraction of phenological markers was done according the same procedure (using Eq.1).

### 2.4. Statistical Analysis

The performance of each of the indirect methods presented above was evaluated with respect to the field phenological observations using three criteria which are (1) the coefficient of determination ($R^2$) calculated from a simple linear regression between estimated ($P_i$) and observed dates ($O_i$) for the different years (*N*), (2) the mean bias error (MBE) and (3) the mean

absolute deviation (MAD) calculated as follows:

$$MBE = \frac{1}{N} \sum_{i=1}^{N} (P_i - O_i)$$

$$MAD = \frac{1}{N} \sum_{i=1}^{N} |(P_i - O_i|$$

## 3. Results

An illustration of time-series of vegetation variables used (OBS, NDVI, NDVI*br*, GCC, (1-*f*R*vis*), *f*APAR, CC, LAI and GPP) is provided in Figure 2. Time-series of all years (2006-2018) are given in the suppl. Fig. S1.

Time-series in Fig. 2 for the year 2015 and in Fig. S1 for all years show that the general patterns of phenological transitions corresponding to the onset of leaves in the spring and to leaf senescence in the autumn are reproduced by all indirect methods but with a variable bias in comparison with the field observation. However, in the autumn, GPP time-series show a decline that appears very early in the year, practically from the beginning of summer. GCC time-series may also show atypical interannual patterns with some years during which a GCC decline, although slower than the one observed on GPP, is also observed very early in the year (2014, 2016-2018 in Fig. S1).

Average phenological dates observed (BB-OBS and LS-OBS) and estimated from the different methods using MOS and MOF markers are given in Figure 3. All phenological dates, using the six phenological markers (SOS, MOS, EOS, SOF, MOF, EOF), are given Table S2.

In spring, field phenological observations (BB-OBS) are earlier than the estimates provided by the majority of the indirect methods (Fig. 3a). However, whatever the method used, the inter-annual phenological variations are well reproduced. During the autumn, phenological observations (LS-OBS) are later than the indirect methods, except for CC and *f*APAR (Fig. 3b), and the performance of the different methods seems more limited compared to spring phenology. Figure 4 shows R², MBE and MAD between observed and estimated phenological dates using MOS (Fig. 4a) and MOF (Fig. 4b) markers during spring and autumnal phenological transitions, respectively.

In the spring, R² values between observed (BB-OBS) and estimated phenological dates (Fig. 4a) based on MOS marker are all statistically significant (at significance level of 0.05) and range from about 0.99 to 0.34. All indirect methods are also consistent with each other as shown by the high correlation coefficients in Fig. S3, which confirms the good reproducibility of interannual phenological variability by the different indirect methods. In comparison to BB-OBS, the best correlation is found with GCC over the period 2012-2018 during which RGB images are available (R² = 0.99). NDVI and CC are also highly correlated with BB-OBS (R² ~ 0.89 and 0.80, respectively). Lower but significant correlations are found between BB-OBS and *f*APAR, LAI, NDVI*br* and 1-*f*R*vis* (R² between 0.6 and 0.7) and the lowest correlation is found between BB-OBS and GPP (R² ~ 0.34).

Between the different indirect methods and during the spring, R² between MOS estimates ranges from 0.26 to 0.96 (see correlation matrix in Fig. S3). Best correlations are found between $f$APAR and NDVI, NDVI$br$, LAI, and $f$R$_{vis}$ (R² >0.89). Good correlations are also found between GCC, NDVI and CC (R² =0.8). Finally, we can also note good consistency between derived dates from GPP- and radiation-based methods (NDVI$br$, $f$APAR, LAI and $f$R$_{vis}$; R² > 0.64). The lowest correlation is found between GCC and GPP.


    For budburst phenological timings, Mean Bias Error (MBE) between BB-OBS and MOS (Fig. 4c) is negative for GCC and CC (estimated date is earlier than observed date). MBE is about -1 day with GCC (MAD ~1 day) over 2012-2018 and is also about -1 day with CC over 2006-2018 (MAD ~ 2 days). We note that MBE or MAD (Figs. 4c and 4e) for these two methods are slightly less than the observation uncertainty of 3.5 days. For the other methods (NDVI, NDVI$br$, LAI, $f$R$_{vis}$,
$f$APAR and GPP) MBE and MAD are equal, meaning that MOS estimates from these methods always overestimate the observed phenological dates BB-OBS. MBE (or MAD) is 3.5 days with NDVI, 6 days with $f$APAR and 8 days with NDVI$br$. MBE is high with LAI (10 days), $f$R$_{vis}$ (14 days) and GPP (17 days). Note that for CC, MBE of about -1 day was obtained after cosine correction of the transmitted PAR according to Eq. 7. Without this correction, MBE increases from -1 day (MAD ~ 2 days) to 6 days (MAD ~ 6 days) and R² decreases from 0.80 to 0.71. Comparison of the phenological patterns of CC time-
series obtained with and without cosine correction shows that the cosine correction has the effect of causing an earlier spring phenological start, thus advancing the date of the inflection point (Fig. S4).

    During the autumn (Fig. 4b), interannual variation of LS-OBS is well reproduced by CC and NDVI time-series which provide estimates that are significantly correlated with the observations (R² =0.80 and 0.63 for CC and NDVI, respectively).
Between the indirect methods (Fig. S3), best correlations are found between NDVI$br$, $f$APAR, NDVI and $f$R$_{vis}$ (R² ~ 0.7), LAI and $f$R$vis$ (R² =0.58), NDVI and $f$APAR (R²=0.56), NDVI and CC (R²=0.55), $f$R$vis$ and $f$APAR (R²=0.55) and CC and $f$APAR (R²=0.42). Surprisingly, correlations between estimated dates from LAI and from CC during the autumn (R² = 0.1), both using the fraction of the transmitted radiation as the unique input, are low compared to what might be expected. Note that it only concerns the senescence stage since the correlation between estimates from LAI and CC during the spring is high (R² ≃ 0.74).
During the senescence phase, for NDVI and CC methods for which the R² between estimates and observations are significant, MBE is of about -2 days with NDVI (MAD ~ 5 days) and about 14 days with CC (MAD ~ 14 days) (Fig. 4d and f). For CC, MBE decreases from about 37 days without cosine correction to 14 days after correction. The cosine correction yields a faster decrease in CC during the senescence stage (Fig. S4). For CC, LS-OBS are better predicted using thresholds at SOF instead of MOF with an MBE of about -1 day (and MAD of 7 days). MOF from LAI, $f$R$_{vis}$, GCC and GPP provide early
estimates compared to LS-OBS. MBE is of about -14 days with LAI, -23 days with $f$R$_{vis}$, -36 days with GCC and -50 days with GPP. $f$APAR leads to estimates that are on average about 30 days later than LS-OBS. Note that for GCC, biases are highly variables between years. For years (2012/2013/2015) for which ADS function does not show the early decline in the autumn, estimated dates are very close to OBS (MBE ~ - 7 days).

For the phenological markers estimated at the beginning and end of budburst (SOS and EOS) or autumn (SOF and EOF) (Table S2), and considering the period 2015-2017 for which the six phenological markers are available from the intensive sampling, it can be noted that SOS dates are close to observed date (DOY 97) for all methods (between DOY 94-101) except for CC. CC starts to increase earlier, at DOY 82, i.e. 15 days before SOS from OBS. Phenological field observations achieved for understory hornbeam trees over the period 2006-2016 (data not shown), show that, on average, the hornbeam budburst date

(i.e. BB-OBS for hornbeam) is around DoY 96 [range 85-107]. MBE between BB-OBS of hornbeam and SOS estimates is about -1 days (MAD ~ 5 days) for GPP, -5 days (MAD ~ 5 days) for NDVI, - 8 days (MAD 8 days) for CC and between 6-8 days for LAI, $f$APAR, NDVI$br$ and $f$R$_{vis}$. For GCC and over 2012-2016, MBE is of 2 days. Significant correlations were also obtained between observed hornbeam budburst dates and SOS estimates derived from NDVI, LAI, NDVI$br$, CC and $f$APAR. R² ranges between 0.73 and 0.49 and the best correlation is obtained with NDVI-based SOS estimates. Note also that there is

a significant correlation between the observed budburst dates of oak and hornbeam (R² ~ 0.6) but on average hornbeam trees break buds about 10 days earlier than oaks.

       For the end of spring, EOS based on GCC are quite close to EOS determined from field phenological observations (3 days earlier for GCC). For the other methods, estimated EOS are later than observed EOS dates. MBE are 3 days for NDVI, 8 days for $f$APAR, 10 days for CC, 14 days for NDVI$br$, 20 days for LAI, 28 days for $f$R$_{vis}$ and 41 days for GPP. During the

senescence phase, SOF from NDVI and CC gives the best agreement with observed SOF date (3 days on average over 2015-2017), followed by $f$APAR (6 days). Observed EOF is better predicted using $f$R$_{vis}$, CC, NDVI and GPP. MBE is about 3 days for $f$R$_{vis}$, 6 days for CC and NDVI and 9 days for GPP.

       As an illustration of the above, Fig. 5 shows average phenological patterns of vegetation variables derived from

average parameters of modelled time-series through ADS function fitted to data over the period 2012-2017, common to all vegetation variables, for the spring (Fig. 5a) and the autumn (Fig. 5b) phenological stages, respectively. The correspondence between field observed dates and phenological metrics derived from indirect methods is also shown.

       Figure 5 illustrates what is described above by showing average temporal patterns during budburst and senescence over the period 2012-2017, common to all eight methods and for which field phenological observations are available in both

spring and autumn. Figure 5a shows the good correspondence between the observed dates and the estimates derived from CC and GCC using the mid-amplitude (50%) MOS threshold. For CC and GCC, MOS clearly marks the budburst date as characterized in the field using the observation protocol used in our study (50% of trees with at least 50% open buds per tree crown, BB-OBS). For the NDVI-based method, on average, the mean observed BB-OBS date coincides with the date when NDVI reaches 25% of its amplitude of variation between NDVI minimum in winter and NDVI maximum at the end of spring.

For the other methods including $f$APAR, NDVI$br$, LAI, $f$R$_{vis}$ and GPP, estimated dates at mid-amplitude threshold are later than BB-OBS with a MAD ranging from 6 to 17 days. A threshold at 20% of the spring amplitude for GPP, $f$R$_{vis}$, NDVI$br$ and at 10% for LAI and $f$APAR provide estimates with a bias < 2 days. During the leaf senescence phase (Fig. 5b), NDVI at mi-

amplitude and CC time-series, just at the start of its decline (~ 95% of its amplitude) provide estimates consistent with the observations. For the other methods, the thresholds shown in Figure 5b are only valid on average over the period 2012-2017 since the relationships between observations and estimates are not statistically significant as shown in Fig. 4b.

Figures 5a and 5b also show that the different methods perform relatively well in the spring but deviate from each other in the autumn. Fig. S5 shows that the relationships between the different variables are dependent on the considered phenological stage. This is clearly the case in the relationships between $f$APAR and NDVI, GCC, GPP, 1-$f$R$_{vis}$. It can be noted that a same NDVI value corresponds to a lower $f$APAR in spring than in autumn. In other words, NDVI and $f$APAR responses to changes in canopy properties follow two different trajectories depending on the season. This "hysteresis" phenomenon may explain the shift between NDVI and $f$APAR-based estimates during the senescence phase (overestimation of the senescence date by the $f$APAR) while both predict very close dates during the spring. This phenomenon of "hysteresis" is also observed in the same way between $f$APAR and GCC or $f$APAR and GPP. A given GPP or GCC value corresponds to a lower $f$APAR in spring than in autumn. We can also note that the relationships between NDVI and GCC are different depending on the season, but for the same NDVI corresponds a higher GCC in spring than in autumn.

## 4. Discussion

### 4.1. Ability of GCC to detect phenological transitions

Using RGB-based GCC (Greenness Chromatic Coordinate index) time-series, the mean absolute deviation (MAD) with BB-OBS is about 1 day over the 7 years of comparison (2012-2018). This result is in line with previous studies, particularly the study of Richardson et al. (2018) who compared RGB-camera based estimates to independent human-eye observations achieved over four deciduous forests. They observed average biases ranging from 1.5 to 6.5 days depending on the site and the best agreement was obtained using GCC at 25% of its amplitude as threshold. Many other studies comparing GCC and indirect visual phenological estimates from same photographs (Klosterman et al. 2014, Wingate et al. 2015) have also concluded that GCC method yields estimations of the spring phenological date with an average bias around 7-8 days. In our study, we show that over the 7-year period (Fig. 5a), GCC at the inflection point (MOS) in spring which corresponds to about 50% of its annual amplitude derived from modelled time-series is the best predictor of the human-eye observed BB-OBS dates which correspond to 50% of sampled oak trees having at least 50% open buds (in fact corresponding to about 50% open buds at the population scale, N. Delpierre unpublished results). This result supports the fact that the camera accurately reports what is observed by human-eye in the field during the spring and that GCC index is a very good indicator of the timing of budburst. It can also be noted that the phenological field observations have been carried out by the same (three) intercalibrated observers over the study period and according to a constant protocol. This may also participate in explaining the good agreement between field observations and estimated dates from RGB-based GCC index time-series. Indeed, several studies have highlighted the importance of uncertainties associated with observations due to various sources, especially

observer effect (Schaber, 2002) and the availability of good quality data is a prerequisite for a rigorous evaluation of the various indirect methods.

On the other hand, the ability of GCC to estimate the senescence date is variable. For some years, the decline in GCC may start earlier than expected, and therefore estimated dates are strongly biased. When the senescence phase causes pronounced contrasts on RGB images between the summer growth and senescence phases, estimated dates agree with field observations, as for the years 2012, 2013 and 2015. For these years, estimated dates are very close to OBS with MAD of about 7 days, of the same order of magnitude as the field observation uncertainty. Therefore, during autumn, data quality and data processing appear crucial to obtain reliable estimates, and extracting of senescence dates based on ADS model may not be the right approach. Other approaches, particularly the spline-based method used for PhenoCam data that has shown good performance (Richardson et al. 2018) deserve to be employed. Other RGB-based spectral indices using the red band, designed specifically to monitor the autumn phenological transition, such as RCC (red chromatic coordinate) (Klosterman et al., 2014; Liu et al. 2020) or GRVI (Green-Red Vegetation Index) (Motohka et al. 2010; Nagai et al. 2012) should also be evaluated. This is beyond the scope of this study and further methodological development is therefore needed to rigorously assess the real potential of this technique for estimating phenological dates during the senescence stage.

Another point to note, as shown in this study (Fig. 2d) and previously pointed in several other studies (Sonnentag et al. 2012; Keenan et al. 2014; Klosterman et al. 2014; Petach et al. 2014) is that GCC shows annual spikes during the spring followed by a rapid decline. The annual amplitude of GCC determined from the modelled time-series is generally smaller than the actual amplitude. In our study, GCC spikes are reached on day 121 on average over 2012-2018. They are not well captured by ADS model because they are delayed by about 10 days compared to the end of spring green-up stage determined from GCC-based EOS (end of spring season) phenological marker. GCC spikes are also reached 10 days before LAI reaches its maximum. This result is consistent with Keenan et al. (2014). Based on intensive field measurements at canopy and leaf scales, they observed a time lag of about two weeks between the canopy maximum LAI measured by LAI-2000 Plant Canopy Analyzer and GCC spikes. They concluded that GCC depends on leaf color and saturates faster than measured canopy LAI, that was explained by the oblique viewing angle of the camera which leads to a higher effective LAI. In the same study, they showed that GCC peaks were reached while main leaf traits (maximum leaf area, chlorophyll content, leaf mass area) continue their development. Similar results were also reported in Yang et al. (2014) and Liu et al. (2015) who showed that GCC peaks in spring were approximately 20 days earlier than the peak of the total chlorophyll concentration. In our study, on average, GCC spikes almost coincide with maximum $f$APAR and CC (EOS) whereas these two variables are based on incoming, reflected and transmitted PAR measurements using hemispherical sensors and therefore are integrative of the whole canopy. This result supports the hypothesis of a combined effect of canopy coloring and closure on GCC spikes. However, and contrary to LAI, which is estimated, in this study, only from incident and transmitted radiation, $f$APAR and CC also additionally use reflected radiation. Therefore, they are also sensitive to changes of leaf color and other leaf traits during the spring. This may explain the good correspondence between the timings of GCC spikes and the timings of maximum of $f$APAR and CC.

#### 4.2. Ability of NDVI to detect phenological transitions

Results also show that MOS and MOF of NDVI are good proxies of observed dates with MAD of about 3-4 days in spring over the whole period 2006-2018 and 5 days in autumn over 2011-2017 period. Estimates based on NDVI are also highly correlated with spring and autumn field phenological observations with an $R^2$ of 0.88 and of 0.62, respectively. This reflects the ability of ground-based NDVI time-series to reproduce the interannual variability of phenology at this site (Figs. 3b and 4b). This potential has also been shown in previous studies, in evergreen and deciduous forest ecosystems in France, an evergreen tropical rain forest in French Guyana, an herbaceous savanna in Congo and a succession of three annual crops in Belgium (Soudani et al. 2012; Hmimina et al. 2013).

Good agreement between RGB-camera indices and proximal NDVI-based measurements has also been shown in crops (Sakamoto et al. 2012) and in herbaceous species (Anderson et al. 2016). However, NDVI measurements does not show the spikes observed on GCC in late spring and our study shows that NDVI is more stable, less scattered, and better representative of LAI plateau throughout the summer growth phase observed in deciduous forests. Similar conclusions were drawn in Petach et al. (2014). In conclusion, the NDVI sensor using MOS and MOF criteria can be considered as the best option since it provides reliable estimates for monitoring both spring and autumn phenology. In addition, and as highlighted in Hmimina et al. (2013), in situ NDVI measurements using proximal sensors are done a few meters above the top of canopy, and because NDVI is a normalized index, the effects of the sky conditions produce little noise. Thus, measurements can be carried out under diffuse sky conditions, allowing for the monitoring of vegetation phenology at high temporal frequency. Nevertheless, proximal NDVI sensors have the disadvantage that measurements remain limited to a narrow field of view and do not allow to extract key phenological metrics at the individual tree level when it may be possible using RGB camera (Delpierre et al. 2020). The use of multispectral cameras with RGB+NIR bands, which are increasingly used on many sites, may allow to overcome this inconvenience and should therefore be encouraged.

#### 4.3. Ability of CC to detect phenological transitions

During the spring, good performance of CC-based method was obtained after cosine correction of the transmitted PAR according to Eq. 7 (Fig. 4a and Table S2). Without this correction, MAD between estimated and observed MOS dates is three times larger (6 days vs 2 days) and $R^2$ slightly lower (0.71 vs 0.80). It can be noted that uncorrected CC, which corresponds to the complement to 1 of the canopy transmittance, and $f$APAR provide similar estimated MOS dates, that are on average about one week later that observed dates (Table S2). This result is in line with the study of Perot et al. (2020), conducted in a mature oak forest, which showed that on average estimated MOS dates from canopy transmittance time-series are about 7 days later than the observed budburst dates.

Comparison of the phenological patterns of CC time-series obtained with and without cosine correction (Fig. S4) shows that the cosine correction has the effect of causing an earlier spring phenological start, thus advancing the date of the

inflection point. While the estimated date at the inflection point after cosine correction (CC-MOS) is very close to BB-OBS, the spring start date (SOS) appears earlier than the observed SOS of oak trees. This can be explained by the budburst of the first trees of the hornbeam understory, which on average has an earlier budburst date, about 10 days before the overstory oak trees. During the senescence phase, the cosine correction significantly improved the estimates, but the bias remains high (14 days on average). Despite this bias, autumn CC-MOF dates are the most correlated with observations LS-OBS ($R^2$ =0.8) (Fig. 4b and Table S2). We notice that CC time-series are sensitive to the intercepted radiation, which mostly depends on canopy structure, and not so much on pigmental (color) properties. Here we derived LS-OBS from the monitoring of the percent of senescent (i.e. colored or fallen leaves) in the canopy, which we build from independent observations of percent colored and percent fallen leaves in the tree crowns. For those years when we continued canopy observations until complete leaf fall, we observed that 50% leaf-fall is typically attained 2-3 weeks after 50%-senescence, at a date comparable to CC-MOF.

In summary, the cosine correction significantly improves estimated dates based on CC both in the spring and senescence seasons. The new formulation of CC calculation proposed in this study (Eq.7), that takes into account the effects of seasonal variations in sun angle on the transmitted PAR, merits being tested at other sites in order to assess accurately its performance as it is likely to be dependent on both the canopy structure and the latitude of the site.

### 4.4. Ability of NDVIbr to detect phenological transitions

The phenological pattern of NDVI*br* is comparable to the one obtained from NDVI time-series but with greater amplitudes during the spring and autumn phenological transitions for the latter (Fig. 2 and Fig. S1). This result is also consistent with Liu et al. (2019) who compared broadband and narrowband NDVI in a temperate broadleaved deciduous forest. Like NDVI, NDVIbr is measured directly above the canopy and seems to be not very sensitive to cloud conditions as also underlined in Wang et al. (2004) and Wilson and Meyers (2007). On average, the deviation between estimated MOS dates from NDVI and NDVI*br* are 5 days in spring and 1 day in autumn, respectively. However, while in spring the estimated MOS dates from NDVI and NDVI*br* are highly correlated ($R^2$ = 0.87), the correlation is lower in autumn ($R^2$=0.49) and is non-significant between autumn NDVI*br* estimates and observed dates LS-OBS. As a result, NDVI and NDVI*br* seem to be decorrelated in autumn and the performance of NDVI*br* time-series to describe the interannual variability of phenology is only limited to spring.

### 4.5. Ability of GPP to detect phenological transitions

On average over an 11-year period (2006-2016), GPP starts increasing (GPP-SOS) on DoY 96, 10 days earlier than overstory oak trees (DoY 106, Fig. 3 and Table S2). The starting date of GPP coincides exactly with the date of hornbeam budburst (DoY 96) and of the earliest oaks (Delpierre et al. 2020). However, GPP reaches a maximum in a time interval close to the summer solstice (Figs. 2 and 5a) and then starts to decline immediately after. Consequently, GPP-MOS overestimates BB-OBS by about 17 days. This result is in line with other previous studies which have shown that GPP peaks several weeks later than the other variables. Toomey et al. (2015) showed that the start of GPP in spring coincides with the onset of GCC,

but GPP peaks 2-4 weeks later. They also noted an immediate decline of GPP once its peak is reached. Similar conclusions between GCC and GPP can also be drawn from Richardson et al. (2009).

During the autumn phase, based on ADS function, the GPP time-series fails to produce plausible estimates of LS-OBS, either using SOF, MOF or EOF criteria.

As underlined above, among all the indirect methods evaluated in this study, estimates of budburst dates derived from GPP time-series using the MOS criterion based on ADS are the most biased estimates and are also the least correlated with the observed phenological dates of oak trees (MBE 17 days, $R^2 = 0.34$, Fig. 4a). This weak correlation can be explained both by a starting of the GPP simultaneously with the budburst of the hornbeam understory and the high dependency of GPP, in addition to the effects of the increase of the LAI and the leaf maturation, to the solar radiation level (Delpierre et al. 2009a). Figs. 2 and 5a show that GPP reaches a short-lived plateau around the summer solstice in June, when both maximum LAI is reached, and solar irradiance is at its maximum. On the other hand, MOF dates during the autumn are earlier than LS-OBS (Figs. 2, 5 and Table S2). Consequently, the length of the period of budburst and leaf development in spring between GPP-derived SOS and EOS dates, is about 57 days over the 13 years of measurements, while it is only about 17 days from *in situ* NDVI. The length of the growing season, between estimated dates of MOS and MOF, is also greatly reduced and it is only 130 days based on GPP, whereas it is 192 days from NDVI and 199 days from field phenological observations. Similar results are shown in the studies of Lu et al. (2018) and Keenan et al. (2014). In conclusion, the extraction of phenology from GPP time-series using inflection points of transitions in the spring and autumn are therefore not representative of the canopy leaf display and other approaches based on absolute or relative thresholds of GPP as in Richardson et al. (2010) and in Wu et al. (2017) may be more representative. Nevertheless, GPP remains a composite signal driven by changes in vegetation phenology and physiological processes that are under the control of the fluctuations of abiotic factors and its use to derive the timings of phenological events must be carried out with great care, as strongly emphasized in Gonsamo et al. (2013).

### 4.6. Hysteresis phenomena between vegetation variables according to the spring and senescence seasons

As shown in Fig. 5, the performance of the different methods for estimating key phenological dates differs between spring and autumn. While the correlations between estimates and observations are all significant during spring (Fig. 4a), only NDVI and CC provide estimates consistent with autumn observations (Fig. 4b). The hysteresis phenomenon that characterizes some relationships between the vegetation variables used in the different methods reflects their different biophysical meanings (Fig. S5). This is particularly the case for the relationships between NDVI and *f*APAR and between GCC and *f*APAR. In spring, the performances of NDVI and *f*APAR are similar, whereas in autumn the *f*PAR provides very late estimates. This can be explained by a high sensitivity of NDVI and GCC to pigment changes during senescence whereas *f*APAR responds mainly to leaf fall and canopy opening.

## 4.7. Linking phenological dates recorded by field phenologists and phenological metrics predicted by indirect proximal methods

The analysis of the link between phenological dates based on field observation and those derived from modelled time-series (Figs. 5a and 5b) shows that, on average over 13 years, BB-OBS (corresponding approximately to 50% buds open in the canopy) are better predicted by MOS (50% of the annual amplitude of variation) for methods based on GCC and CC. For NDVI-based method, a threshold of 25% of its amplitude coincides with the average observed date. However, due to the rapid increase of NDVI during the spring, a 50% threshold also provides estimates with a bias of the same order of magnitude as the uncertainty in the phenological observations (3.5 days). For the other methods (GPP, $f$R$_{vis}$, NDVI$br$, $f$APAR and LAI), a threshold at 20% of the annual amplitude appears more appropriate to estimate the average observed date of budburst. During the senescence phase, and for NDVI- and CC-based methods, for which observations and estimates are significantly correlated, MOF of NDVI is very close to the observed LS-OBS date (50% of trees having at least 50% of senescent or fallen leaves per tree crown) and SOF of CC is more in line with the observed date but less stable than MOF.

Although they are based on data acquired over a long period covering 13 years of measurements and observations, these thresholds may be specific to our study site and their stability and genericity merit further study in other forest ecosystems.

## 4.8. Summary remarks on deriving phenological metrics from radiation-based methods in EC flux-tower sites

Many EC flux-tower sites that use the eddy covariance technique routinely acquire the biometeorological variables used in the calculation of GPP, LAI, $f$R$_{vis}$, NDVI$br$, $f$APAR and CC. During the spring stage, LAI, $f$R$_{vis}$ and GPP-based estimates are biased by about 10 to 17 days. $f$R$_{vis}$ and GPP are the worst performing predictors, especially GPP. On the other hand, this study shows that NDVI$br$, $f$APAR and CC are able to reproduce interannual variation of spring budburst with a bias of about one week when MOS is considered (Figs. 3 and 4, Table S2). In same vein, the use of CC based-method is also another robust alternative for monitoring spring and autumn phenological transitions in EC flux-tower sites. However, CC and $f$APAR require additional measurements of transmitted radiation below the canopy. Indeed, such measurements are not commonly achieved at EC flux measurement sites and should be deployed as, in addition to phenology, transmitted radiation data time-series can also be used to estimate Leaf Area Index and to characterize its seasonal dynamics (Keenan et al. 2014). These measurements must be performed using an appropriate number of below-canopy radiation sensors to take the heterogeneity of the canopy structure into account (Pontailler, 1990; Link et al. 2004; Garrity et al. 2011; Webster et al. 2016). When such data are available, derived phenological metrics can be used to conduct retrospective studies in order to interpret the interannual variability of carbon fluxes and are preferable to those derived from the fluxes themselves such GPP or NEP, as already pointed in Gonsamo et al. (2013).

## 5. Conclusion

We used various methods to characterize the temporal dynamics of forest canopy in a temperate deciduous forest. Field phenological observations provided exhaustive multi-year samples allowing to accurately assess the potential of each method. However, we emphasize that this potential remains relative because it was evaluated using ADS method applied to all vegetation proxies considered in this study as the only method of extracting phenological dates in order not to bias their comparison. Using ADS-based phenology extraction method, results show that this potential is different depending on the method and the season. During the spring phase, GCC, NDVI and CC, using the inflection point MOS criterion, provide estimates closest to observed dates with an absolute bias less than 4 days, of the same order as the temporal resolution of phenological observations (3.5 days). For CC, this is obtained only after a cosine correction of the transmitted PAR, correction that takes the variation of the optical path in the canopy due to the seasonal variation of the solar angle into account. Without this correction, the prediction bias increases from about 2 days to 6 days. Using MOS criterion, NDVI$br$ and $f$APAR give also satisfactory estimates with a bias around one week that corresponds to the temporal resolution generally used in phenological observations. However, for these variables as well as for $f$R$_{vis}$, LAI and GPP, a threshold of 20% of their transition amplitude in spring allows to obtain more precise estimates in agreement with observed dates. During the senescence phase, only MOF of NDVI and CC can reproduce the interannual variability of leaf senescence. However, these findings are specific to the ADS-based method used to derive phenological markers from time-series data. More appropriate methods, especially for GPP and GCC time series, could have provided better estimates of senescence date.

This study validated the estimates provided by the different methods by comparing them with phenological observations carried out according to the same protocol by intercalibrated observers and over 13 years of field observations for budburst and 7 years for leaf senescence. But more particularly, this study demonstrated the good performance of methods based on broad band NDVI (NDVI$br$), the fraction of absorbed PAR ($f$APAR) and canopy closure (CC) that use solar radiation data routinely recorded at several EC flux tower sites. This opens real perspectives to conduct retrospective studies for a better interpretation of the interannual variation of carbon fluxes. $f$APAR and CC use transmitted radiation measurements below the canopy which are less common but merit being largely deployed at EC flux measurement sites.

*Author contribution:* KS designed the study. He did the data analysis and wrote the manuscript with contributions from all co-authors. ND, DB, JYP, LS, GV and ED participated in the data collection and its formatting.

*Competing interests.* The authors declare that they have no conflict of interest.

*Acknowledgments.* Many thanks to all colleagues who participated in the installation of the various instruments on the Fontainebleau-Barbeau site, and all those involved in the data collection used in this study. The FR-Fon study site has been funded through several French and European research framework programmes (GIP Ecofor, Allenvi, CarboEurope, FP6;

CarboExtreme, FP7). It is part of the Integrated Carbon Observation System (ICOS, FP7) European research infrastructure, and of the SOERE-Ecofor French research network.

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

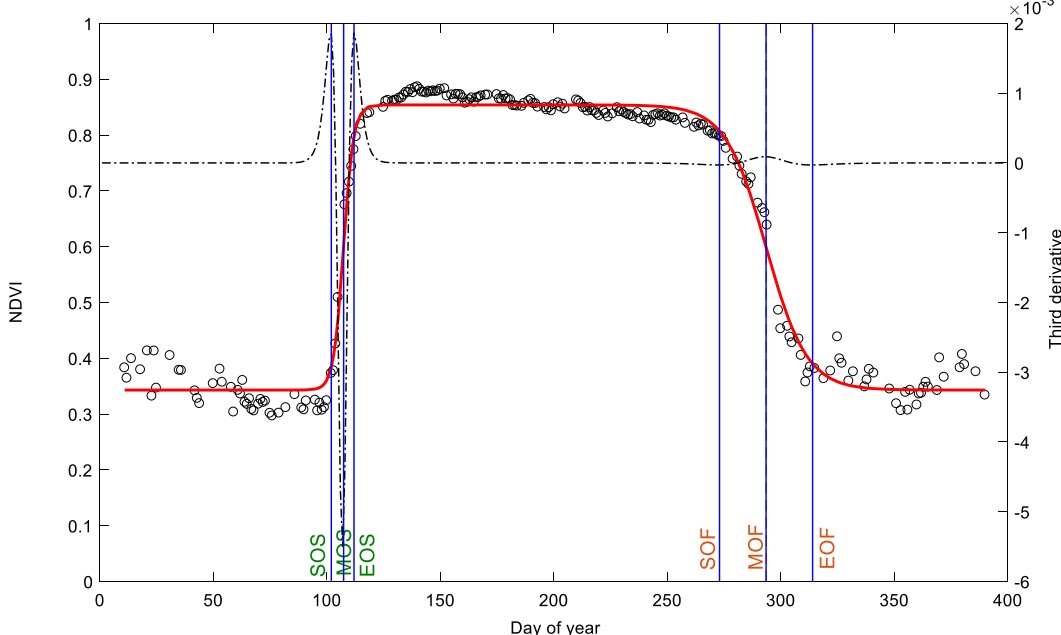

**Figure 1**: Illustration of phenological markers extracted from ADS (Asymmetric double sigmoid) functions fitted to NDVI data acquired in 2015 (empty circle and red curve). Vertical lines in blue: SOS, MOS and EOS are dates of start, middle and end of leaf onset in spring. SOF, MOF and EOF are dates of start, middle and end of leaf senescence (colored and fallen leaves) in autumn. The third derivative of the ADS function showing peaks and holes corresponding to the six phenological dates (black dotted line).

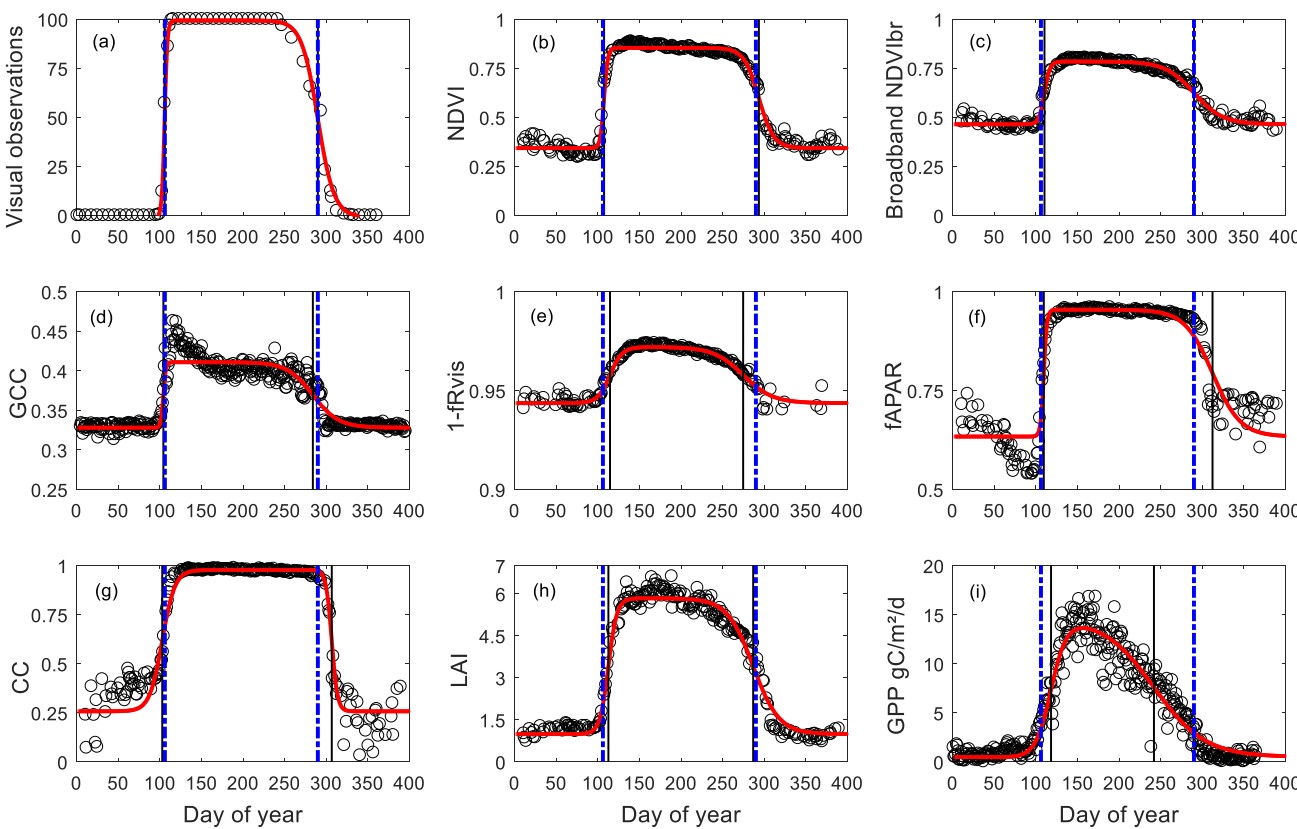

**Figure 2**: Illustration of one-year (2015) time-series of OBS (a), NDVI (b), NDVI*br* (c), GCC (d), 1-*f*R*vis* (e), *f*APAR (f), CC (g), LAI (h) and GPP (i) in Fontainebleau-Barbeau forest. Data are shown in empty circle. The red bold continuous curve is the ADS function (Eq. 1) fitted to time-series. For visual observations, data shown are in % of open buds in spring and in % of non-senescent leaves (100% – observed percentage of senescent leaves) in autumn. % of open buds is forced to 100% for the summer growing season ant to 0% during the winter dormancy season. Vertical lines: spring and autumn phenology estimates using MOS and MOF (black) and observed dates (BB-OBS and LS-OBS) (blue).

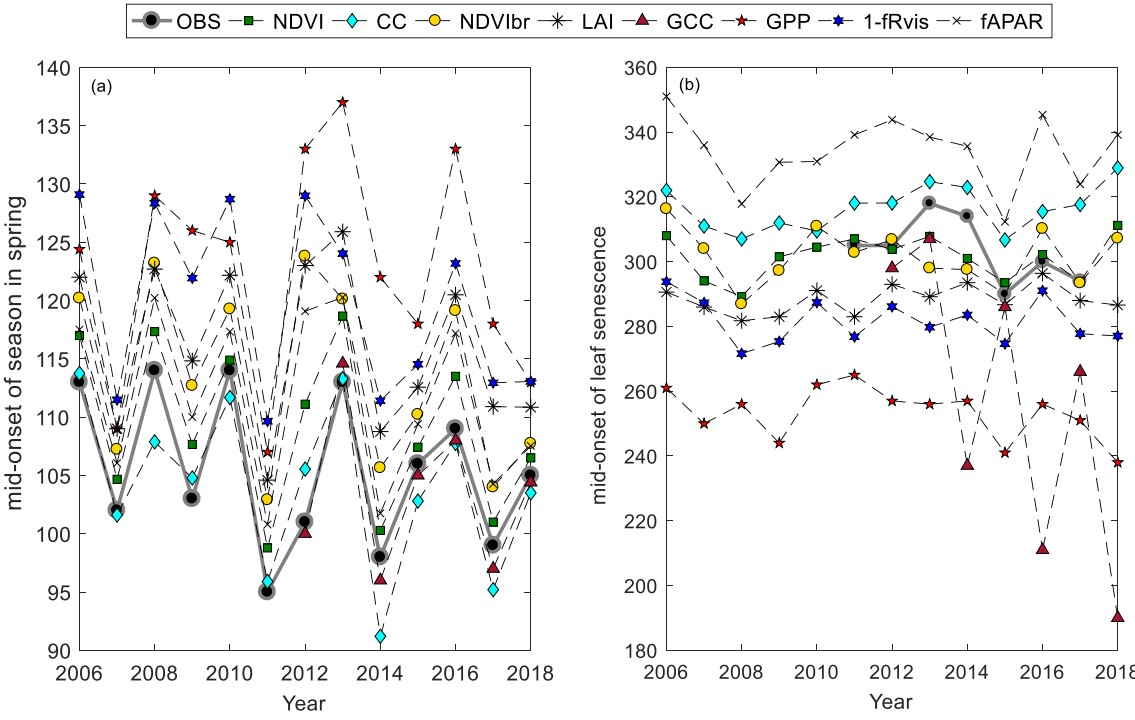

**Figure 3**: Average phenological dates in spring (a) and autumn (b) using MOS and MOF phenological markers, respectively, and for the different years.

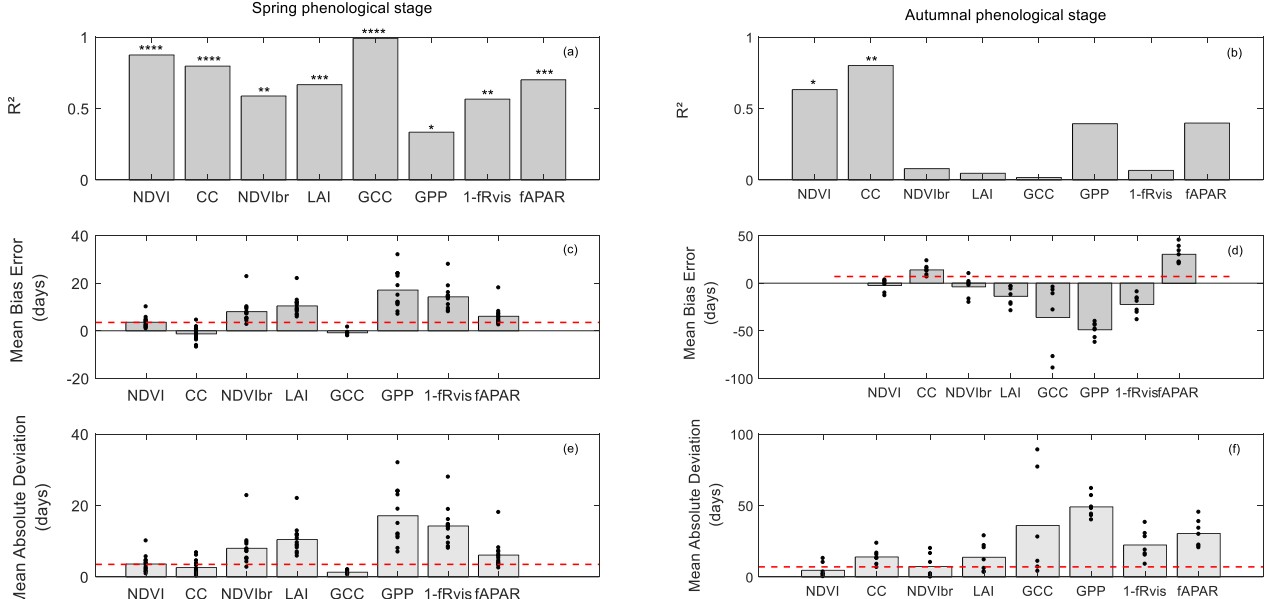

**Figure 4**: Coefficient of determination (R²) (a and b), mean bias error (MBE) (c and d) and mean absolute deviation (MAD) (e and f) in days between observed and estimated phenological dates using MOS and MOF markers during spring (a, c and e) and autumnal (b, d and f) phenological stages. The significance levels of R² are given by stars: * $P <0.05$, ** $P < 0.01$, *** $P < 0.001$ and **** $P < 0.0001$. The height of grey boxes marks the average of the statistics across study years (individual years are represented by the black dots). Red horizontal lines represent temporal-resolution related uncertainties associated with field phenological observations of 3.5 days during the spring and of 7 days during the autumn.

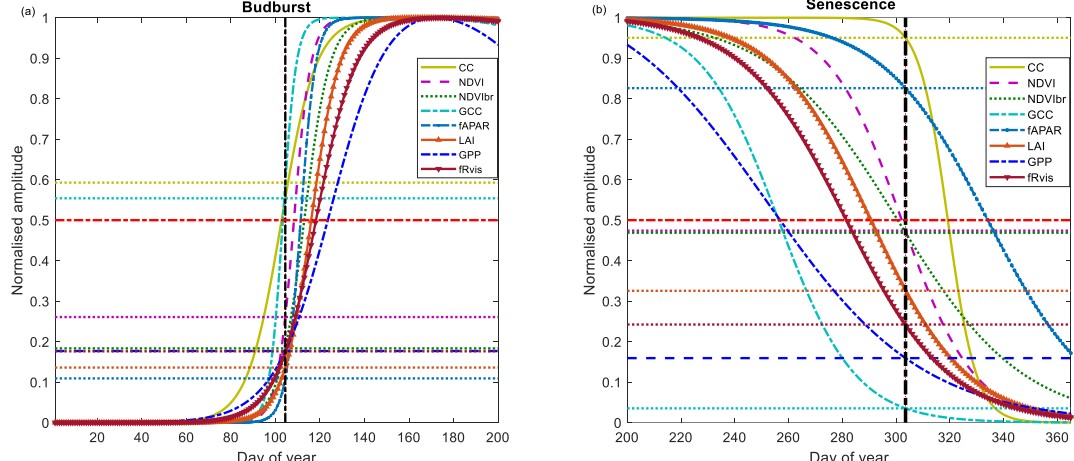

**Figure 5**: Average phenological patterns during budburst (a) and senescence (b) during the period 2012-2017 using modelled time-series through ADS function fitted on the measured time-series of NDVI (Normalized Difference Vegetation Index), GCC (Greenness Chromatic Coordinate), broad-band NDVI (NDVI*br*), LAI (Leaf Area Index), *f*APAR, CC (Canopy Closure), *f*R*vis* (fraction of reflected radiation) and GPP (Gross Primary Production). Amplitudes of variations are normalized to 1. Horizontal dotted lines: for each variable, proportion of the average amplitude that equals the average of the BB-OBS (Fig. 5a) and LS-OBS (Fig. 5b) dates. Horizontal bold red line (y-axis = 0.5): mid-amplitude (50%) corresponding to mid-onset of spring (MOS) and mid-onset of senescence (MOF). Vertical black line: averages of observed phenological dates during 2012-2017 for budburst (BB-OBS) and for senescence (LS-OBS).

**Table 1.** Methods and variables used in the calculation of phenology metrics in the Fontainebleau-Barbeau Forest. NDVI: narrow-band normalized difference vegetation index; NDVI*br*: broad-band NDVI; *f*R*vis*: fraction of reflected radiation by the canopy in PAR spectral domain; GCC: greenness chromatic coordinate from RGB camera images; *f*APAR: fraction of absorbed radiation in PAR spectral domain; CC: canopy closure; LAI: leaf area index; GPP: Gross Primary Productivity. These vegetation variables are named $V_v$ hereafter.

| Method ($V_v$) | Data used to calculate Vv | Period | Time resolution |
|---|---|---|---|
| Human-eye phenological observations (OBS) | % open buds (spring) % senescent (colored or fallen) leaves (autumn) | 2006-2018 (spring) 2011-2015; 2015-2017 (autumn)* | Twice a week (spring) Once a week (autumn) |
| GCC index | AXIS-Camera RGB images | 2012-2018 | Hourly (8-17 h UT) |
| Narrow-band NDVI | Radiances in red and near infrared bands | 2006-2018 | Half hourly |
| Broad-band NDVI*br* | Incoming and reflected radiation in PAR and shortwave spectral regions | 2006-2018 | Half hourly |
| *f*R*vis* | Fraction of reflected radiation in PAR spectral region | 2006-2018 | Half hourly |
| Fraction of absorbed PAR (*f*APAR) | Incoming, reflected and below-canopy transmitted radiation in PAR spectral region | 2006-2018 | Half hourly |
| Canopy closure (CC) | Incoming and below-canopy transmitted radiation in PAR spectral region | 2006-2018 | Half hourly |
| Leaf Area index (LAI) | Incoming and below-canopy transmitted radiation in PAR spectral region | 2006-2018 | Half hourly |
| Gross Primary Productivity (GPP) | Gross $CO_2$ assimilation by the ecosystem, calculated from eddy covariance data | 2006-2018 | Half hourly |

\* see text for details