# Peer review of "A survey of proximal methods for monitoring leaf phenology in temperate deciduous forests"

_Biogeosciences, 2020_

## Referee Comment (RC1) · Anonymous Referee #1 · 9 Jan 2021

General comment:

The study is designed carefully, and has an exhaustive discussion covering a range of relevant previous studies. There are no major concerns from my side, except a few specific comments listed below.

Line comments:

L9-10: But there have been some (e.g., PhenoCam network; Milliman et al. 2019).

L25-26: There are ongoing debates on how such a temperature-driven control has been changed, and about other factors controlling vegetation phenology like photoperiod and chilling requirements.

L34: How about satellite-based observations? Now their spatial coverages span 3 to 500 meters, and some of them have over 20 to 40 year-long records covering the entire globe.

L100-104: The dates derived from the extrema of the third derivative are quite comparable with the dates from amplitude thresholds. However, these are not identical, and their relationships depend on the rate of increase/decrease in vegetation index during growing/senescence phase.

L343: For Fig. S5, could it be possible to show the relationships of OBS with others? That would be interesting.

References

Milliman, T., Seyednasrollah, B., Young, A.M., Hufkens, K., Friedl, M.A., Frolking, S., Richardson, A.D., Abraha, M., Allen, D.W., Apple, M. and Arain, M.A., 2019. Pheno-Cam Dataset v2. 0: Digital Camera Imagery from the PhenoCam Network, 2000–2018. ORNL Distributed Active Archive Center.

---

## Referee Comment (RC2) · Anonymous Referee #2 · 17 Mar 2021

This study is very well written and the results, which compare phenological dates derived from multiple vegetation proxies using a double sigmoidal function to ground observations at a forest site in France, are clearly presented. The comparison of the dates derived from multiple proxies shows the strengths and weaknesses of each, at least when using a double sigmoid model.

My main concern is that the results are generally specific to the use of a double sigmoid function, and could change significantly if methods tailored to individual proxies were applied. In addition, each data source (camera imagery, photosynthesis estimates, radiometric) is taken 'as-is' and little quality control of the underlying time series is

applied. The results are not presented as thus however, and instead are presented as a quantification of the underlying information content of the examined proxies. I suggest the authors either reconsider the framing of the manuscript as an assessment of the potential of using one approach applied to multiple and varied proxies with little quality control, rather than presenting the results as a direct assessment of the proxies examined.

Detailed comments: Line 10: the term 'flux measurement site' is ambiguous. I assume you mean eddy-covariance (EC) flux measurement sites. Note that tree phenology is recorded at many (if not a lot) EC sites.

Line 20: 'GPP provides the most biased estimates' should read 'our method to derive phenological dates from GPP provides the most biased estimates' or something along these lines (perhaps: "the assymetic double sigmoidal function we used to derive phenological dates provides the most biased estimates for GPP"). The point being that the results are both method and data dependent, and better estimates could in theory be derived from GPP using a method more suited to noisy time series.

Lines 87-107: Applying a '1 model fits all' approach to estimating the phenological thresholds is questionable. For instance, a lot of effort has gone into estimating robust transition dates for Gcc, including multiple signal processing steps to improve the signal to noise ratio and the fitting of multi-dimensional splines for date estimation (see e.g., Richardson et al. 2018 (https://www.nature.com/articles/s41598-018-23804-6) which performs a similar study as that presented here though with many more sites and fewer vegetation proxies). This is particularly important for GCC, as the seasonal cycle in deciduous forests does not follow a smooth double-logistic curve (which is why your model has trouble fitting it, e.g., in Fig. 2d). The same point applies to many of the other data sources considered. For example for GPP, detailed outlier detection and other quality control efforts are needed (see, e.g., https://essd.copernicus.org/preprints/essd-2020-58/essd-2020-58.pdf, or the PhAsT framework: https://www.nature.com/articles/nclimate2253). This is particularly important for GPP given that it is subject to relatively large random error and shows high variability due to changes in the environment. It is hard therefore to determine whether the comparison results can be meaningfully interpreted. How much of the difference between proxies is due to the fact that the method you applied is less suitable for a particular proxy? Without detailed, proxy-specific, data processing based on the state of the art it is certain that your statistical characterization of the utility of each proxy considerably underestimates its true potential.

Line 246 Figure 2: It is very hard to distinguish the data from the ADS model here. Convention suggests using black for the data and red for the model. It is important for the reader to be able to clearly see the fitted model in order to assess the robustness of the derived dates.

Figure 3: This should be in color. It is very difficult to see which line corresponds to which data source.

---

## Author Comment (AC1) · 7 Apr 2021

We are very grateful to the two referees for their constructive criticism, which helped to improve the manuscript. Please find below our responses point by point to comments and questions and a detailed account of the changes made to the first version of the manuscript.

- General comment: The study is designed carefully, and has an exhaustive discussion covering a range of relevant previous studies. There are no major concerns from my side, except a few specific comments listed below.

We are pleased with this positive feedback on our work.

- comments: L9-10: But there have been some (e.g., PhenoCam network; Milliman et al. 2019). Milliman, T., Seyednasrollah, B., Young, A.M., Hufkens, K., Friedl, M.A., Frolking, S., Richardson, A.D., Abraha, M., Allen, D.W., Apple, M. and Arain, M.A., 2019. PhenoCam Dataset v2. 0: Digital Camera Imagery from the PhenoCam Network, 2000–2018. ORNL Distributed Active Archive Center.

Thank you for this remark. Our sentence was not clear. We wanted to emphasize that the monitoring of phenology on EC flux sites was not systematic at their beginning, although it is a very important variable to interpret the temporal variability of fluxes and carbon stocks in the concerned ecosystems. Indeed, the phenocam network, which started in the early 2000s, was the first to implement routine monitoring of phenology at carbon flux measurement sites in the USA through standards protocols of image acquisition and extraction of phenological dates. This strongly stimulated the installation of similar networks in Europe (http://european-webcam-network.net/) and other countries (Australia, Japan for examples). We have referred to Phenocam network in our introduction in the first version of our manuscript, citing in particular the papers of Richardson, 2019; Klosterman et al. 2014 and Sonnentag et al. 2012. In this version, we added the paper by Richardson and colleagues (2018) which present the dataset documented in Milliman et al. (2019). In this version, we explicitly refer to the data acquired within the Phenocam framework by citing the Richardson et al. 2018 and Milliman et al. 2019.

Richardson, A.D., Hufkens, K., Milliman, T., Aubrecht, D.M., Chen, M., Gray, J.M., Johnston, M.R., Keenan, T.F., Klosterman, S.T., Kosmala, M., Melaas, E.K., Friedl, M.A., Frolking, S., 2018. Tracking vegetation phenology across diverse North American biomes using PhenoCam imagery. Sci Data 5, 1–24.

Milliman, T., Seyednasrollah, B., Young, A.M., Hufkens, K., Friedl, M.A., Frolking, S., Richardson, A.D., Abraha, M., Allen, D.W., Apple, M. and Arain,

M.A. et al., 2019. PhenoCam Dataset v2.0: Digital Camera Imagery from the PhenoCam Network, 2000-2018. ORNL DAAC, Oak Ridge, Tennessee, USA. https://doi.org/10.3334/ORNLDAAC/1689

We also have modified L9-L10 as follows:

" Yet, tree phenology has rarely been monitored in a consistent way throughout the life of a flux tower site."

- L25-26: There are ongoing debates on how such a temperature-driven control has been changed, and about other factors controlling vegetation phenology like photo-period and chilling requirements.

We fully agree with the reviewer's comment. This is the first sentence of the introduction and we just wish here to introduce the subject by remembering the prominent effect of temperature, not entering the details of the control of leaf phenology.

- L34: How about satellite-based observations? Now their spatial coverages span 3 to 500

To avoid needlessly burdening the text, we have deliberately chosen not to refer to spatial remote sensing because we have focused our study on in situ methods. Nevertheless, we have cited in this manuscript our studies on vegetation phenological metrics extraction using satellite data time-series (Soudani et al. 2018 and Hmimina et al. 2013).

- L100-104: The dates derived from the extrema of the third derivative are quite comparable with the dates from amplitude thresholds. However, these are not identical, and their relationships depend on the rate of increase/decrease in vegetation index during growing/senescence phase.

We agree with this remark. Indeed, the dates at 10% and 90% are not identical to the extrema of the third derivative. For the numerically determined 10% and 90% during spring and fall phenological transitions, there is indeed a small shift. During spring

phase, the 10% date comes slightly later than the first maximum of the third derivative and the 90% date is slightly earlier than the second maximum of the third derivative. During the fall, 10% date during the decay phase is later than the date determined from the third derivative (first minimum) and the 90% decay date is slightly earlier. We agree that the difference depends on the rate of change in vegetation index. However, the 10% and 90% phenological stages remain interesting because they span the spring and winter transition phases, but the determination of the corresponding dates is less robust than the date at the inflection point. We have changed the text as follows:

L100: "For these two dates u and v, Vv(t) is very close to 50% of its total amplitude of variation, in spring and autumn respectively."

L103-104: "SOS, MOS and EOS for the start, middle, and end of leaf onset (budburst) in spring and SOF, MOF and EOF for the start, middle and end of leaf senescence in autumn, corresponding approximately to 10%, 50% and 90% of total amplitude during the increase and the decline in canopy greenness in spring and autumn, respectively."

- L343: For Fig. S5, could it be possible to show the relationships of OBS with others? That

It is not possible to establish the same type of relationship between OBS and the other variables. These relationships are established from daily measurements. Observations are made twice a week during the spring and once a week during the fall.

---

## Author Comment (AC2) · 7 Apr 2021

We are very grateful to the two referees for their constructive criticism, which helped to improve the manuscript. Please find below our responses point by point to comments and questions and a detailed account of the changes made to the first version of the manuscript.

- This study is very well written and the results, which compare phenological dates derived from multiple vegetation proxies using a double sigmoidal function to ground observations at a forest site in France, are clearly presented. The comparison of the

dates derived from multiple proxies shows the strengths and weaknesses of each, at least when using a double sigmoid model. My main concern is that the results are generally specific to the use of a double sigmoid function and could change significantly if methods tailored to individual proxies were applied.

We agree with the reviewer and thank her/him for this insightful comment. Indeed, the estimation of phenological dates from the different vegetation proxies used can be achieved by different approaches and by choosing different phenological indicators. The potential of each proxy is also dependent on the method and data used. In this study, our first objective is to compare the different proxies taking care to minimize as much as possible the methodological biases. We have therefore chosen to apply the same processing protocol for the determination of phenological dates. We have chosen to use the double sigmoid and to extract the six indicators because : this method is increasingly used and can be applied to all vegetation proxies used in our study; this method is very suitable for extracting phenological dates from time series with high temporal resolution; the six phenological markers determined are well defined allowing us to undertake a careful comparison of the different proxies considering not only the two phenological dates of spring and autumn but also the entire phenological transition of spring and autumn by analyzing also the dates at 10% and 90%. Our results can also be compared to other studies and throughout the analysis of our results and the discussion we tried to take into account the possible biases associated with the method and the data (see lines 380-387 about the RGB-based method and lines 487-493 in the first and the revised versions of our manuscript). From our perspective, one of the major difficulties encountered in studies of phenology from vegetation proxies is the lack of a standard method and indicators that allow comparison of different studies. It is clear that the multitude of methods and indicators in the literature reflects the fact that there is no true "best method" because the quality of the estimates is mostly dependent on the data used. In this study, we show that each proxy taken separately can provide "credible" estimates of key phenological dates, but significant differences appear when comparing the different proxies. We wanted to highlight these differences

by comparing the vegetation proxies to each other and to estimates from direct in situ observations. Without the use of the same method of determining phenological dates, the comparison of the potential of the different proxies will be difficult. But, we agree, that the potential of each proxy may be better if other more appropriate methods are applied. This is what we wanted to emphasize in L380-387 about the RGB-Based method and in lines 487-493 about the GPP-based method.

We changed the text as follows:

L93-95: "Then, to compare the different vegetation proxies without possible methodological biases, we opted for the same method using an asymmetric double sigmoid (ADS) similar to Zhang et al. (2003); Soudani et al. (2008); Klosterman et al. (2014)."

L380-L387: "Therefore, during autumn, data quality and data processing appear crucial to obtain reliable estimates, and extracting of senescence dates based on ADS model may not be the right approach. Other approaches, particularly the spline-based method used for PhenoCam data that has shown good performance (Richardson et al. 2018) deserve to be employed. Other RGB-based spectral indices using the red band, designed specifically to monitor the autumn phenological transition, such as RCC (red chromatic coordinate) (Klosterman et al., 2014; Liu et al. 2020) or GRVI (Green-Red Vegetation Index) (Motohka et al. 2010; Nagai et al. 2012) should also be evaluated. This is beyond the scope of this study and further methodological development is therefore needed to rigorously assess the real potential of this technique for estimating phenological dates during the senescence stage".

- In addition, each data source (camera imagery, photosynthesis estimates, radiometric) is taken 'as-is' and little quality control of the underlying time series is applied. The results are not presented as thus however, and instead are presented as a quantification of the underlying information content of the examined proxies. I suggest the authors either reconsider the framing of the manuscript as an assessment of the potential of using one approach applied to multiple and varied proxies with little quality control,

rather than presenting the results as a direct assessment of the proxies examined.

With respect to data quality control, we do not fully agree with the reviewer. The time series used were finely analyzed and a rigorous noise removal procedure was applied. To minimize the effects of sky conditions on the NDVI signals and vegetation indices calculated from the radiation measurements, we limited the use of the data to the 10 am - 2 pm measurement interval and applied a thresholding method described in Soudani et al. 2012 (L154-158). In this manuscript, we also proposed a simple method to minimize the effects of seasonal variations of sun angle on transmitted PAR (L204). All-time series are shown in the Appendix S1. Regarding the Greenness Chromatic Coordinate (GCC) from RGB images, GCC time series do not contain significant noise because GCC was calculated from pseudo-reflectance in Green, Red and Blue by standardizing reflected radiance by the radiances measured on a white target (L160-170). We also considered the daily average value to constitute GCC time series. But as underlined above, the problem does not seem to be related to data quality, but to the ADS method which seems to not be well adapted to extract phenological dates from GCC time-series, especially during the senescence phase.

Detailed comments: -L10: the term 'flux measurement site' is ambiguous. I assume you mean eddy-covariance (EC) flux measurement sites. Note that tree phenology is recorded at many (if not a lot) EC sites.

We have changed L9-10 and in several places in the manuscript: L9-L10: " Yet, tree phenology has rarely been monitored in a consistent way throughout the life of a flux tower site."

- L20: 'GPP provides the most biased estimates should read 'our method to derive phenological dates from GPP provides the most biased estimates' or something along these lines (perhaps: "the assymetic double sigmoidal function we used to derive phenological dates provides the most biased estimates for GPP"). The point being that the results are both method and data dependent, and better estimates could in theory be

derived from GPP using a method more suited to noisy time series.

We completely agree with this remark. Thank you very much for the suggestion. This paragraph has been modified as follows: L20: "The double asymmetric sigmoid function (ADS) we used to derive the phenological dates provides the most biased estimates for the GPP". We have also taken into account this remark in line 474.

- Lines 87-107: Applying a '1 model fits all' approach to estimating the phenological thresholds is questionable. For instance, a lot of effort has gone into estimating robust transition dates for Gcc, including multiple signal processing steps to improve the signal to noise ratio and the fitting of multi-dimensional splines for date estimation (see e.g., Richardson et al. 2018 (https://www.nature.com/articles/s41598- 018-23804-6) which performs a similar study as that presented here though with many more sites and fewer vegetation proxies). This is particularly important for GCC, as the seasonal cycle in deciduous forests does not follow a smooth double logistic curve (which is why your model has trouble fitting it, e.g., in Fig. 2d).

We totally agree with this remark. We already highlighted this issue in the first version (lines 380-387 in the revised version). We have added the reference to the work of Richardson et al. 2018 (see our response to the first comment). Richardson A.D., Hufkens K., Milliman T., Frolking S. 2018. Intercomparison of phenological transition dates derived from the PhenoCam Dataset V1.0 and MODIS satellite remote sensing. Sci. Rep., 8 (2018), p. 5679. doi.org/10.1038/s41598-018-23804-6

- The same point applies to many of the other data sources considered. For example for GPP, detailed outlier detection and other quality control efforts are needed (see, e.g., https://essd.copernicus.org/preprints/essd-2020-58/essd-2020-58.pdf, or the PhAsT framework: https://www.nature.com/articles/nclimate2253). This is particularly important for GPP given that it is subject to relatively large random error and shows high variability due to changes in the environment. It is hard therefore to determine whether the comparison results can be meaningfully interpreted. How much of the difference

between proxies is due to the fact that the method you applied is less suitable for a particular proxy? Without detailed, proxy-specific, data processing based on the state of the art it is certain that your statistical characterization of the utility of each proxy considerably underestimates its true potential.

We agree with all these remarks.

For the GPP, we think that the problem is not so much the existence of outliers because our GPP time series are relatively clean (see appendix S1) but that the extraction method based on the sigmoid is not well adapted in particular for the extraction of the senescence date. We highlighted this point in the first draft and in the revised version of the manuscript.

L487-493: "In conclusion, the extraction of phenology from GPP time-series using inflection points of transitions in the spring and autumn are therefore not representative of the canopy leaf display and other approaches based on absolute or relative thresholds of GPP as in Richardson et al. (2010) and in Wu et al. (2017) may be more representative."

Indeed, each proxy can potentially provide better estimates with a more adapted method. In our study, we are aware of this problem, but we preferred to use the same method for comparison. Our conclusions are indeed more or less method specific. However, the ADS-based method is a common method used to derive phenological markers from time-series data both using in situ data and spatial remote sensing. As explained in our response to the first comment, the aim of this work isn't to compare or improve phenological metrics per se, but to compare vegetation proxies within a consistent framework. As such, we opted for one standard method. While the optimization of phenological metrics extraction would indeed be highly interesting, in our case, it would require significant efforts to not only optimize this extraction for each vegetation proxy independently, but also do ensure that each vegetation proxy benefits from the same level of optimization. Such task would be highly challenging, and would be outside the scope of this article. We have modified our conclusion as follows: L533- 538: "We used various methods to characterize the temporal dynamics of forest canopy in a temperate deciduous forest. Field phenological observations provided exhaustive multi-year samples allowing to accurately assess the potential of each method. However, we emphasize that this potential remains relative because it was evaluated using ADS method applied to all vegetation proxies considered in this study as the only method of extracting phenological dates in order not to bias their comparison. Using ADS-based phenology extraction method, results show that this potential is different depending on the method and the season."

L546-548: "However, these findings are specific to the ADS-based method used to derive phenological markers from time-series data. More appropriate methods, especially for GPP and GCC time series, could have provided better estimates of senescence date."

- Line 246 Figure 2: It is very hard to distinguish the data from the ADS model here. Convention suggests using black for the data and red for the model. It is important for the reader to be able to clearly see the fitted model in order to assess the robustness of the derived dates.

- Figure 3: This should be in color. It is very difficult to see which line corresponds to which data source.

Both figures 2 & 3 have been changed according to your suggestions.